# Intestinal Microbiome Associated with Efficacy of Atezolizumab and Bevacizumab Therapy for Hepatocellular Carcinoma

**DOI:** 10.3390/cancers16091675

**Published:** 2024-04-26

**Authors:** Yosuke Inukai, Kenta Yamamoto, Takashi Honda, Shinya Yokoyama, Takanori Ito, Norihiro Imai, Yoji Ishizu, Masanao Nakamura, Masatoshi Ishigami, Hiroki Kawashima

**Affiliations:** Department of Gastroenterology and Hepatology, Nagoya University Graduate School of Medicine, 65 Tsurumai-cho, Showa-ku, Nagoya 466-8560, Japany-ishizu@med.nagoya-u.ac.jp (Y.I.); makamura@med.nagoya-u.ac.jp (M.N.);

**Keywords:** microbiome, hepatocellular carcinoma, immune checkpoint inhibitor, atezolizumab, bevacizumab

## Abstract

**Simple Summary:**

The combination of atezolizumab and bevacizumab is a standard treatment for unresectable hepatocellular carcinoma. This study investigated the relationship between the gut microbiome and treatment efficacy. Fecal samples from 37 patients with hepatocellular carcinoma treated with this combination were analyzed. Patients were divided into responders (*n* = 28) and non-responders (*n* = 9). While overall microbiome diversity was similar, certain bacteria, such as *Bacteroides stercoris* and *Parabacteroides merdae* were more abundant in responders. Patients lacking these bacteria had worse prognoses. This suggests that differences in gut microbiota play a role in the effectiveness of atezolizumab and bevacizumab therapy.

**Abstract:**

The combination of atezolizumab and bevacizumab has become the first-line treatment for patients with unresectable hepatocellular carcinoma (HCC). However, no studies have reported on specific intestinal microbiota associated with the efficacy of atezolizumab and bevacizumab. In this study, we analyzed fecal samples collected before treatment to investigate the relationship between the intestinal microbiome and the efficacy of atezolizumab and bevacizumab. A total of 37 patients with advanced HCC who were treated with atezolizumab and bevacizumab were enrolled. Fecal samples were collected from the patients, and they were divided into responder (*n* = 28) and non-responder (*n* = 9) groups. We compared the intestinal microbiota of the two groups and analyzed the intestinal bacteria associated with prognosis using QIIME2. The alpha and beta diversities were not significantly different between both groups, and the proportion of microbiota was similar. The relative abundance of *Bacteroides stercoris* and *Parabacteroides merdae* was higher in the responder group than in the non-responder group. When the prognosis was analyzed by the presence or absence of those bacteria, patients without both had a significantly poorer prognosis. Differences in intestinal microbiome are involved in the therapeutic effect of atezolizumab and bevacizumab.

## 1. Introduction

Hepatocellular carcinoma (HCC) is a leading cause of death worldwide, despite advancements in Hepatitis B virus (HBV) and Hepatitis C virus (HCV) treatments [1]. Various drugs, including lenvatinib, regorafenib, ramucirumab, and cabozantinib, have been developed for unresectable HCC [2,3,4,5]. However, atezolizumab and bevacizumab, a combination of immune checkpoint inhibitors (ICIs) and angiogenesis inhibitors, have been shown to be more effective than molecular target agents and are widely used as a first-line treatment [6].

ICIs are thought to have an indirect anti-tumor effect by inducing T-cell activation mechanisms [7]. The administration of ICIs augments T-cell-mediated immune responses against tumor cells and has been found to improve overall survival in patients with various cancer types [8,9]. However, the response to these treatments varies, and markers for treatment efficacy have not yet been identified [10]. Several studies have shown that the gut microbiome influences the antitumor immune response and that modulation of the microbiome may improve therapeutic response [11,12]. Thus, the gut microbiome is thought to be closely related to the function of ICIs.

The liver and the intestinal tract are connected through the portal vein, and as the gut–liver axis, it is often reported to be closely related to the progression of liver disease and the carcinogenesis and progression of HCC [13,14]. There have been many reports of differences in specific bacteria, and recently Jun et al. reported that Ruminococcaceae, Porphyromonadaceae, and Bacteroidetes are associated with a decreased risk of HCC [15]. Although several reports have been published on the association between ICI-treatment for HCC and the gut microbiome [16,17], no reports have investigated whether the gut microbiome is directly related to the efficacy of atezolizumab and bevacizumab for HCC. Therefore, this study aimed to investigate the relationship between the efficacy of the combination therapy of atezolizumab and bevacizumab in unresectable HCC and the gut microbiome by analyzing the gut microbiome of patients prior to treatment.

## 2. Materials and Methods

### 2.1. Study Design

This study was designed as a single-center cohort study to investigate the relationship between the gut microbiome and the efficacy of atezolizumab and bevacizumab therapy in patients with unresectable HCC. Patients who received this combination therapy at Nagoya University Hospital between December 2020 and October 2022 and who provided written informed consent were included in the study.

### 2.2. Treatment Response and Patient Groups

Treatment response was assessed using the Response Evaluation Criteria in Solid Tumors (RECIST) v1.1 [18] during the treatment period, with responses classified as complete response (CR), partial response (PR), stable disease (SD), or progressive disease (PD). Treatment efficacy was evaluated every 6 to 8 weeks from the start of treatment, and patients were divided into two groups based on the best response achieved during treatment: a responder group with treatment response classified as either SD, PR, or CR and a non-responder group with no treatment response or PD.

### 2.3. Survival Analysis

Overall survival (OS) was defined as the time from the first day of treatment to the date of death or last follow-up. Progression-free survival (PFS) was defined as the time from the first day of treatment to the date of PD onset or death.

### 2.4. Ethics Approval

This study was conducted in accordance with the Declaration of Helsinki and approved by the ethics committee of Nagoya University Hospital (approval number: 2015-0420). All patients provided written informed consent before enrollment.

### 2.5. Patients

Patient data, including sex, age, body mass index (BMI), cause of chronic hepatitis, blood test data, previous treatment, use of proton pump inhibitors (PPIs), and antibiotics use in the month prior to administration were collected from medical records. HCC stage was classified using the Barcelona Clinic Liver Cancer Classification (BCLC) [19], and liver function was classified using the Child–Pugh score [20]. Diabetes mellitus was diagnosed in accordance with the criteria of the American Diabetes Association, with a random plasma glucose level of ≥200 mg/dL, fasting plasma glucose level of ≥126 mg/dL, or use of any anti-hyperglycemic medication [21]. Alcohol consumption of more than 80 g/day was considered heavy drinking [22].

Before treatment, patients were screened for varices by endoscopy to assess the risk of bleeding. Patients with high-risk varices received endoscopic treatment before administration of atezolizumab and bevacizumab. In addition, patients were screened for endocrine, neovascular, and respiratory abnormalities before administration of combination therapy.

### 2.6. Treatment Protocol

Patients received a combination therapy of atezolizumab and bevacizumab (Chugai Pharmaceutical Co., Ltd., Tokyo, Japan) via intravenous administration. The dosage consisted of 1200 mg of atezolizumab and 15 mg/kg of bevacizumab, given every 3 weeks until either tumor progression or unacceptable adverse events occurred. If clinical benefits were observed, treatment was continued even beyond tumor progression. In the event of adverse events, patients were allowed to receive monotherapy of either atezolizumab or bevacizumab, depending on the type of adverse event experienced.

### 2.7. Sample Collection and DNA Isolation

Sample collection and DNA isolation were performed as previously reported [23]. In brief, fecal samples were collected from patients before treatment and stored at −80 °C. DNA was extracted using the DNeasy PowerSoil Kit (Qiagen, Hilden, Germany) and stored at −80 °C until further analysis. The V3-4 regions of the bacterial 16S rRNA gene were amplified using universal primers, and the PCR products were purified with AMPure XP magnetic beads (Beckman Coulter, Brea, CA, USA). Samples were barcoded and pooled for sequencing on an Illumina MiSeq platform using the MiSeq Reagent Kit v3 with 2 × 300 reads and 600 cycles (Illumina, San Diego, CA, USA).

### 2.8. 16S rRNA Gene Sequencing

The 16S rRNA sequencing data was processed using QIIME2 (version 2021.2) [24]. After demultiplexing, the paired-end reads for 37 microbiome samples were imported into QIIME2. Sequence quality control and feature table construction was performed using the Divisive Amplicon Denoising Algorithm 2 (DADA2) QIIME2 plugin. After denoising, a pre-trained naive Bayes classifier based on the SILVA database version 132 was used to explore the taxonomic distribution of the samples [25].

### 2.9. Statistical Analyses

Statistical analyses were conducted using EZR, a graphical user interface for R [26]. The significance level was set at *p* < 0.05. Continuous variables were expressed as medians with interquartile ranges and analyzed using the Mann–Whitney U test. Categorical variables were analyzed using chi-squared or Fisher’s exact tests. Microbiome data were analyzed and visualized using the MicrobiomeAnalyst online platform [27]. Alpha diversity was calculated using the Chao 1, Shannon index, and observed genuses, and beta diversity was estimated using the Bray–Curtis index and visualized using principal coordinate analysis (PCoA). The significance of the PCoA plot was determined using PERMANOVA. Differential taxonomy analysis was performed using linear discriminant analysis effect size (LEfSe: “http://huttenhower.sph.harvard.edu/galaxy/” accessed on 1 April 2023) with LDA score > 2 and *p* value < 0.05 [28]. Survival analysis was performed using the Kaplan–Meier method, and differences between groups were assessed by log-rank test.

## 3. Results

### 3.1. Patient Background

During the study period, a total of 54 patients received the combination therapy of atezolizumab and bevacizumab at our hospital. Out of these, 37 patients agreed to participate in the study and provided pre-treatment fecal samples. Of these 37 patients, 28 were categorized as responders (11 patients with PR and 17 patients with SD) and 9 were categorized as non-responders (Figure 1).

Patient background information for both groups is presented in Table 1.

The median age of patients in the responder and non-responder groups was 74 and 75 years, respectively (*p* = 0.184). The sex ratio (male:female) was 23:5 in the responder group and 5:4 in the non-responder group (*p* = 0.178). The etiology of HCC was as follows: HBV (5 patients), HCV (7 patients), alcohol (10 patients), and non-B non-C (6 patients) in the responder group and HBV (1 patient), HCV (6 patients), alcohol (2 patients), and non-B non-C (0 patients) in the non-responder group (*p* = 0.128). The distribution of BCLC stage was as follows: A (3 patients), B (6 patients), and C (19 patients) in the responder group and A (0 patients), B (3 patients), and C (6 patients) in the non-responder group (*p* = 0.707). The distribution of Child–Pugh scores was as follows: 5 (22 patients), 6 (6 patients), and 7 (0 patients) in the responder group and 5 (3 patients), 6 (3 patients), and 7 (3 patients) in the non-responder group (*p* = 0.007), indicating significantly better liver function in the responder group. During the month prior to treatment, there were three patients in the responder group who had used antibiotics: the first patient was on erythromycin for bronchiectasis, the second patient took amoxicillin for approximately 10 days due to pyelonephritis, and the third patient regularly used rifaximin due to hyperammonemia. In the non-responder group, there was one patient who used amoxicillin for three days due to dental treatment. Neutrophil-to-lymphocyte ratio (NLR), which has been reported to be associated with the efficacy of ICIs, did not differ between the two groups [29]. The presence of varices before the administration of atezolizumab and bevacizumab was observed in 11 patients in the responder group and 3 individuals in the non-responder group. Preventive endoscopic treatment prior to administration was administered to one patient in each group. No patients experienced difficulties in treatment continuation due to variceal rupture during the administration period. Other blood test data and the use of PPIs and antibiotics were similar between the two groups.

### 3.2. Microbiome Profiling and Comparison of Alpha and Beta Diversities

We examined the gut microbiome profile of the responder and non-responder groups at the phylum level, as shown in Figure 2.

Both groups were dominated by the major phyla found in human gut microbiota: Firmicutes, Bacteroidetes, Actinobacteria, and Proteobacteria. The proportions of each phylum were as follows: Firmicutes (78.0% vs. 68.9%), Bacteroidetes (19.7% vs. 14.1%), Actinobacteria (5.3% vs. 8.3%), and Proteobacteria (2.5% vs. 3.1%) in the responder and non-responder groups, respectively. We also compared the alpha and beta diversities of the two groups and found no significant differences (Appendix A).

### 3.3. Comparison of Relative Abundance of Bacterial Communities between Responder and Non-Responder Groups with LEfSe

We used LEfSe to compare the relative abundance of bacterial communities between the responder and non-responder groups to identify bacterial genera associated with treatment response. Figure 3 shows the results of the LEfSe analysis, with green and red bars indicating bacterial communities that were significantly increased in the responder and non-responder groups, respectively.

The graph shows a comparison of the relative abundance of bacterial communities between both groups performed with LEfSe. Green bars show bacterial communities with a higher relative abundance in the responder group, and red bars represent bacterial communities that were more abundant in the non-responder group. LEfSe, Linear Discriminant Analysis Effect Size.

One family (*Tannerellaceae*), one genus (*Parabacteroides*), and two species (*Bacteroides stercoris* and *Parabacteroides merdae*) were found to be more abundant in the responder group. Conversely, the microbiome related to Phylum Synergistota and three families (*Leuconostocaceae, Morganellaceae*, and *Synergistaceae*), two genera (*Weisella* and *Lachnospiraceae UCG 008*), and some species were enriched in the non-responder group.

### 3.4. Comparison of Progression-Free Survival (PFS) and Overall Survival (OS)

To investigate the bacteria associated with the effect of atezolizumab and bevacizumab, we focused on *Bacteroides stercoris* and *Parabacteroides merdae*, which were enriched in the responder group and have been reported to be associated with a better effect of ICIs [30,31]. The relative abundance of these bacteria in each sample is shown in Appendix A. PFS and OS were calculated for the two groups based on the presence or absence of each bacterium (Appendix A). Both bacteria showed a trend towards better PFS and OS in the group with the bacterium, and particularly the group with *Bacteroides stercoris* showed significantly better PFS than the group without the bacteria.

On the basis of these results, we divided the patients into four groups based on the presence or absence of the bacteria (1: *P. merdae*(−) + *B. stercoris*(−), 2: *P. merdae*(+) + *B. stercoris*(−), 3: *P. merdae*(-) + *B. stercoris*(+), 4: *P. merdae*(+) + *B. stercoris*(+)). PFS and OS were analyzed separately. The results showed that the group that was negative for both bacteria had significantly worse PFS and OS than the other groups (Appendix A). Comparison of patient backgrounds in each group showed that the group without both bacteria had significantly higher alanine aminotransferase (ALT) and α-fetoprotein (AFP) (Appendix A). Because the absence of both bacteria may be related to the poor efficacy of atezolizumab and bevacizumab treatment, we analyzed OS in the group negative for both bacteria (*n* = 4) and in the group positive for at least one of either bacterium (*n* = 33). We found that the group without the bacteria had a significantly worse prognosis (Figure 4).

The upper graphs show a comparison of PFS and OS in the group with and without *Bacteroides stercoris*. PFS was significantly longer in the group with *Bacteroides stercoris*, and OS tended to be longer. The lower graphs show a comparison of PFS and OS in the group with and without *Parabacteroides merdae*. Both tended to be longer in the group with *Parabacteroides merdae*.

PFS, Progression Free Survival; OS, Overall Survival; *B. stercoris*, *Bacteroides stercoris*; *P. merdae*, *Parabacteroides merdae*.

Comparison of patient background showed that the group members with the absence of these two bacteria were significantly older, with lower serum albumin and higher AFP levels (Table 2).

## 4. Discussion

In this study, we analyzed pre-treatment fecal samples from patients undergoing atezolizumab and bevacizumab treatment and identified bacteria enriched in the responder group that were associated with better prognosis.

The recent introduction of several new agents has led to advancements in the treatment of HCC. Atezolizumab, an ICI, is commonly used in combination with bevacizumab, an angiogenesis inhibitor, as a first-line treatment for unresectable HCC. However, the efficacy of this treatment and potential adverse events remain unclear.

Through the “gut-liver axis”, a bidirectional communication pathway between the gut and the liver, intestinal bacteria, and their metabolites exert profound effects on liver health. Dysbiosis, an imbalance in the gut microbiota composition, is associated with various liver conditions, such as metabolic dysfunction-associated steatotic liver disease, alcoholic liver disease, and ultimately, HCC [13,14].

Many studies have reported differences in gut microbiota in patients with HCC in animal and clinical studies, and research is underway to determine the mechanisms underlying these differences [32,33]. Several key metabolites produced by intestinal bacteria influence liver function and disease progression. These include short-chain fatty acids, bile acids, and lipopolysaccharides [34]. Dysbiosis can disrupt the production and balance of these metabolites, leading to detrimental effects on liver metabolism and immune response regulation.

The chronic inflammation and liver fibrosis that often precede HCC development are promoted by dysbiosis-induced inflammation and genotoxicity. These factors can lead to DNA damage, genomic instability, and dysregulated cell proliferation and apoptosis pathways, all of which are involved in hepatocarcinogenesis [14].

In recent years, numerous studies have reported on the relationship between ICIs and the intestinal microbiome, with intestinal bacteria-mediated mechanisms thought to be involved in their efficacy and in adverse events [11,12]. In relation to HCC, a study from Taiwan revealed that *Lachnoclostridium* and *Prevotella 9* are associated with the favorable effects of ICI administration in HCC [17]. Meanwhile, another study from Japan showed that the efficacy of atezolizumab and bevacizumab was related to the use of antibiotics, suggesting that intestinal bacteria play a role in the effectiveness of these drugs [35]. In our study, *Parabacteroides merdae* and *Bacteroides stercoris* were found to be enriched in the responder group. Both of these bacteria have been reported to have favorable effects on ICIs [30,31]. *Parabacteroides merdae* has been reported to potentiate the effects of ICI by being involved in decreasing regulatory T cells, increasing the frequency of Batf3-linage dendritic cells and greater T helper cell 1 responses [36]. In a study comparing the combination treatment of anti-Programmed cell Death 1 (PD-1) therapy and with anti-PD-1 therapy alone in a mouse model of colorectal cancer, the combination treatment was more effective than anti-PD-1 therapy alone, and *Bacteroides stercoris* was one of the intestinal bacteria involved in the metabolic changes in the combination group, with a higher abundance in the combination group [37]. Therefore, it is possible that in our study, these bacteria may have been involved in both the immune function and metabolic products of the patients and increased the effect of atezolizumab and bevacizumab.

The absence of both *Bacteroides stercoris* and *Parabacteroides merdae* was significantly associated with poor prognosis. Comparing baseline characteristics of these patients, the group without either bacterium was significantly older (83 vs. 73 years) and had higher AFP. Even among the patients aged 75 or older, AFP was significantly higher in the group without both bacteria. AFP levels at the time of HCC diagnosis have been reported as an independent risk predictor related to pathological grade, progression, and survival [38]. This suggests that the presence or absence of these bacteria may indicate patients with higher risk of malignancy.

Our study had some limitations. Owing to the small number of cases in this single-center, retrospective study, differences between the microbiome and treatment effects by etiology or liver function were difficult to examine. Further studies with larger patient cohorts or validation groups are needed to address confounding factors and interfering factors related to the gut microbiome (e.g., race, alcohol intake, quality of defecation, dietary habits). Moreover, metabolite measurements were not taken, and thus the mechanism by which the microbiome may be involved in the effects of ICIs is unknown.

## 5. Conclusions

Our study suggests that specific gut bacteria may play a crucial role in determining the treatment response of HCC patients receiving atezolizumab and bevacizumab therapy. These findings may pave the way for the development of novel microbiome-based therapeutic strategies for enhancing the efficacy of immunotherapy in HCC patients.

## Figures and Tables

**Figure 1 cancers-16-01675-f001:**
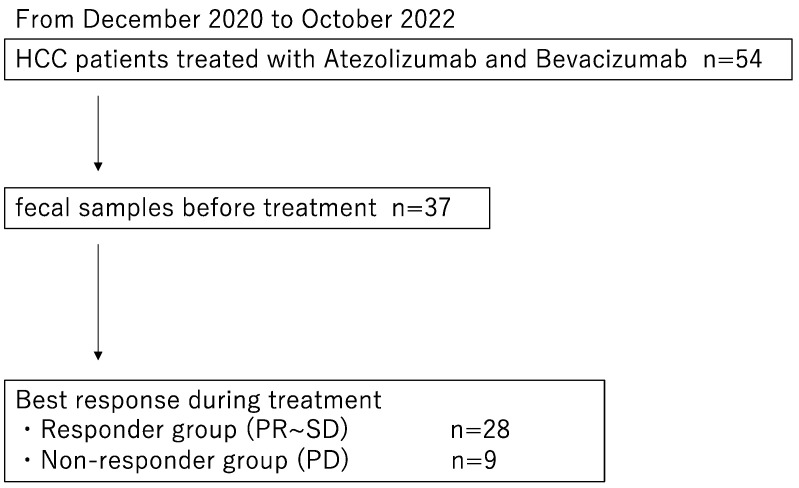
Study design. HCC, hepatocellular carcinoma; PR, Partial Response; SD, Stable Disease; PD, Progressive Disease.

**Figure 2 cancers-16-01675-f002:**
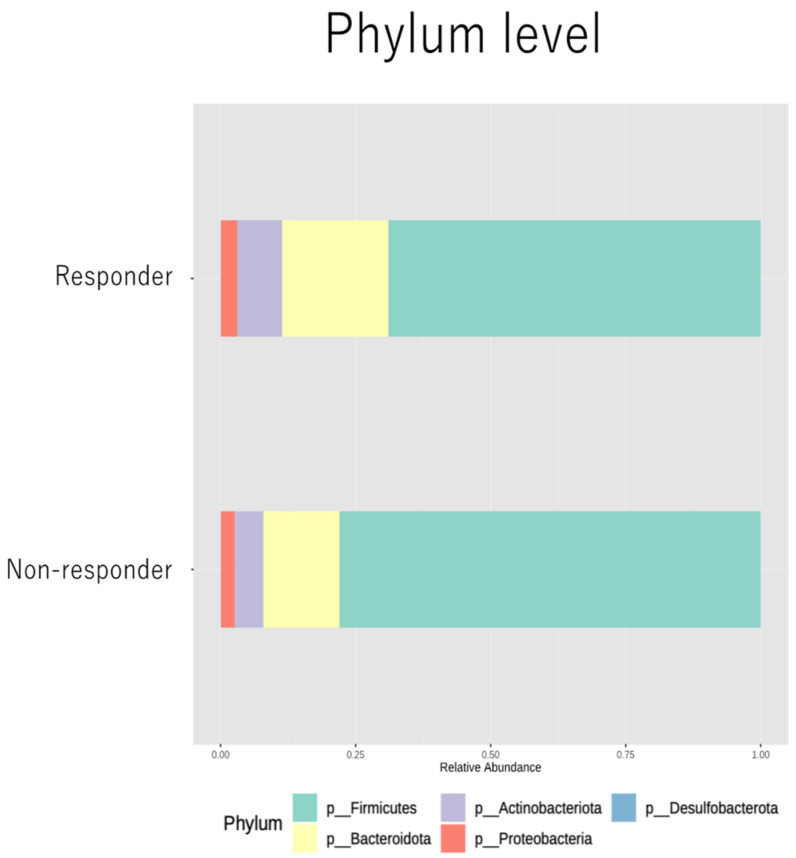
Relative abundances at phylum level in responder and non-responder groups.

**Figure 3 cancers-16-01675-f003:**
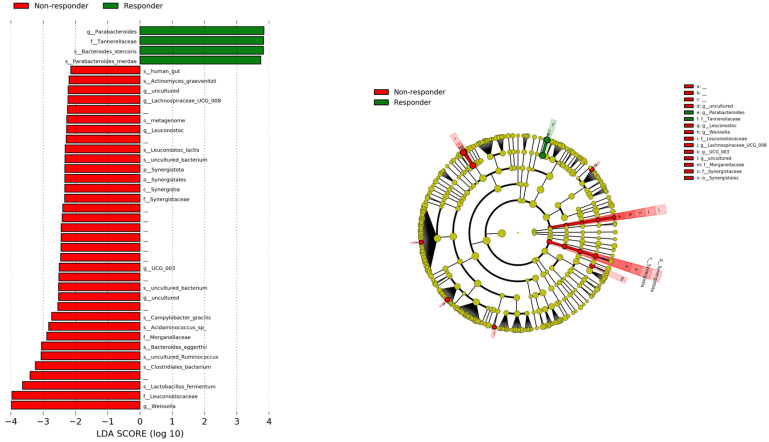
Comparison of relative abundance of bacterial communities between groups performed with LEfSe.

**Figure 4 cancers-16-01675-f004:**
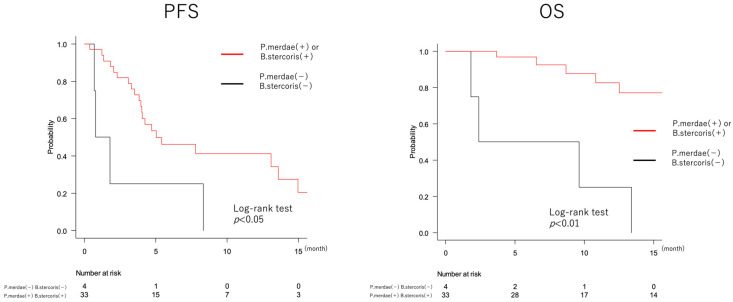
Comparison of Progression Free Survival and Overall Survival grouped by presence or absence of *Bacteroides stercoris* and *Parabacteroides merdae*.

**Table 1 cancers-16-01675-t001:** Patient characteristics.

	Responder *n* = 28	Non-Responder *n* = 9	*p*-Value
Age †	74 (63–79)	75 (71–85)	0.184
Gender (male/female)	23/5	5/4	0.178
Body mass index †	23.3 (20.9–26.8)	24.0 (20.9–25.3)	0.804
Etiology (HBV/HCV/Alcohol/NBNC)	5/7/10/6	1/6/2/0	0.128
BCLC stage (A/B/C)	3/6/19	0/3/6	0.707
Child–Pugh grade (A/B)	28/0	6/3	0.011
Child–Pugh score (5/6/7)	22/6/0	3/3/3	0.007
Treatment history (1/2/3/4)	26/2/0/0	4/3/0/2	0.005
AST (IU/L) †	36 (27–51)	37 (20–46)	0.710
ALT (IU/L) †	29 (21–38)	25 (15–30)	0.279
γ-GTP (IU/L) †	77 (55–136)	33 (31–186)	0.583
Total bilirubin (mg/dL) †	0.8 (0.7–1.1)	0.7 (0.5–1.0)	0.246
Albumin (g/dL) †	3.8 (3.6–4.0)	3.4 (2.6–3.8)	0.051
HbA1c (%) †	6.1 (5.6–6.8)	6.2 (5.7–6.7)	0.657
AFP (ng/mL) †	9.5 (5.0–410.8)	564.0 (13.0–845.0)	0.123
PPI (yes/no)	18/10	5/4	0.705
Antibiotics (yes/no)	3/25	1/8	1.000
C-reactive protein (mg/dL) †	0.20 (0.09–0.50)	0.38 (0.04–1.46)	0.645
NLR †	1.96 (1.29–3.32)	2.71 (1.64–3.62)	0.357

BCLC, the Barcelona Clinic Liver Cancer Classification; AST, aspartate aminotransferase; ALT, alanine aminotransferase; γ-GTP, γ-glutamyl transpeptidase; AFP, α-fetprotein; PPI, protom pump inhibitor; NLR, Neutrophil–Lymphocyte Ratio. † Values are expressed as median (interquartile range).

**Table 2 cancers-16-01675-t002:** Patient background with and without *P. merdae* and *B. stercoris*.

	*P. merdae* (+) or *B. stercoris* (+) *n* = 33	*P. merdae* (−) and *B. stercoris* (−) *n* = 4	*p*-Value
Age †	73 (63–79)	83 (79–85)	0.047
Gender (male/female)	26/7	2/2	0.244
Body mass index †	24.0 (20.9–27.0)	21.7 (20.1–23.3)	0.240
Etiology (HBV/HCV/Alcohol/NBNC)	6/10/11/6	0/3/1/0	0.460
BCLCstage (A/B/C)	3/8/22	0/1/3	1
Child–Pugh grade (A/B)	31/2	3/1	0.298
Child–Pugh score (5/6/7)	24/7/2	1/2/1	0.104
Treatment history (1/2/3/4)	27/5/1	3/0/1	0.278
AST (IU/L) †	35 (25–45)	134 (88–174)	0.106
ALT (IU/L) †	27 (21–35))	52 (23–107)	0.365
γ-GTP (IU/L) †	72 (34–132)	266 (147–447)	0.149
Total bilirubin (mg/dL) †	0.8 (0.6–1.2)	0.9 (0.7–1.0)	0.825
Albumin (g/dL) †	3.8 (3.6–4.0)	3.0 (2.8–3.3)	0.039
HbA1c (%) †	6.1 (5.7–6.8)	6.2 (5.9–6.8)	0.980
AFP (ng/mL) †	10.0 (5.0–494.0)	4335.5 (429.8–36,901.0)	0.027
PPI (yes/no)	21/12	2/2	0.625
Antibiotics (yes/no)	4/29	0/4	1.000
C-reactive protein (mg/dL) †	0.20 (0.09–0.44)	1.27 (0.82–1.47)	0.186
NLR †	1.97 (1.22–2.75)	3.65 (3.20–4.08)	0.070

BCLC, the Barcelona Clinic Liver Cancer Classification; AST, aspartate aminotransferase; ALT, alanine aminotransferase; γ-GTP, γ-glutamyl transpeptidase; AFP, α-fetprotein; PPI, protom pump inhibitor; NLR, Neutrophil–Lymphocyte Ratio. † Values are expressed as median (interquartile range).

## Data Availability

The data that support the findings of this study are available from the corresponding author upon reasonable request.

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
