# Peer review of "Intestinal Microbiome Associated with Efficacy of Atezolizumab and Bevacizumab Therapy for Hepatocellular Carcinoma"

_cancers, 2024, doi:10.3390/cancers16091675_

Round 1
Reviewer 1 Report
Comments and Suggestions for Authors
Thank you for sharing your research. Here are some suggestions for improvement:
1. The authors reported a single-center observational study conducted in 28 responder group patients and 9 non-responder group patients, with a relatively small sample size that may not effectively support the conclusion that differences in the gut microbiome are related to the treatment effects of Atezolizumab and Bevacizumab. It would be beneficial to supplement the number of patients enrolled in each group.
2. The primary etiology of HCC patients significantly differed between the responder and non-responder groups. Should this factor be considered when discussing the prognostic differences between the two groups?
3. The study only mentioned whether the enrolled patients used antibiotics but did not specify the type and duration of antibiotics nor whether other confounding factors, such as gastrointestinal infections, could affect the gut microbiome results in the current study.
4. The authors mentioned that some patients in the study underwent endoscopic treatment for varices. However, there is no specific data or related discussion on the number of patients who underwent endoscopic therapy in each group, and this difference could affect the overall survival between groups explored in the study.
5. It is recommended to supplement the references to illustrate the research achievements in this field in recent years. Additionally, in light of the results of this study, it would be beneficial to analyze the current possible confounding or interfering factors.
Comments on the Quality of English LanguageMinor editing of English language required.
Author Response
Thank you for sharing your research. Here are some suggestions for improvement:
- The authors reported a single-center observational study conducted in 28 responder group patients and 9 non-responder group patients, with a relatively small sample size that may not effectively support the conclusion that differences in the gut microbiome are related to the treatment effects of Atezolizumab and Bevacizumab. It would be beneficial to supplement the number of patients enrolled in each group.
Thank you for this important comment. With great regret, it is difficult to collect additional samples in the limited time available. Furthermore, the sample size was similar to ours in the recent report shown below that examined the effects of ICI on hepatocellular carcinoma and intestinal microbiome. Despite the small sample size, we believe there is great value in reporting our study because studies using atezolizumab plus bevacizumab as an ICI in this setting are still rare, even with this sample size. Therefore, we are considering reporting on this sample size for the time being.
Lee PC, Wu CJ, Hung YW, et al. Gut microbiota and metabolites associate with outcomes of immune checkpoint inhibitor-treated unresectable hepatocellular carcinoma. J Immunother Cancer. 2022;10(6):e004779. doi:10.1136/jitc-2022-004779
Shen YC, Lee PC, Kuo YL, et al. An Exploratory Study for the Association of Gut Microbiome with Efficacy of Immune Checkpoint Inhibitor in Patients with Hepatocellular Carcinoma. J Hepatocell Carcinoma. 2021;8:809-822. Published 2021 Jul 24. doi:10.2147/JHC.S315696
- The primary etiology of HCC patients significantly differed between the responder and non-responder groups. Should this factor be considered when discussing the prognostic differences between the two groups?
Thank you for pointing out this concern. There used to be reports of differences in the effectiveness of ICI depending on etiology. However, recent investigations have shown no association with etiology. Therefore, it is believed that etiology did not influence the treatment efficacy in this case.
Espinoza M, Muquith M, Lim M, Zhu H, Singal AG, Hsiehchen D. Disease etiology and outcomes after atezolizumab plus bevacizumab in hepatocellular carcinoma: post-hoc analysis of IMbrave150. Gastroenterology. 2023; 165(1): 286-288.e4.
Copil FD, Campani C, Lequoy M, et al. No correlation between MASLD and poor outcome of Atezolizumab-Bevacizumab therapy in patients with advanced HCC. Liver Int. 2024;44(4):931-943. doi:10.1111/liv.15833
- The study only mentioned whether the enrolled patients used antibiotics but did not specify the type and duration of antibiotics nor whether other confounding factors, such as gastrointestinal infections, could affect the gut microbiome results in the current study.
Thank you for pointing out this concern. We apologize for not providing a breakdown of antibiotic users. Among the four individuals who used antibiotics, within the responder group, one of the three individuals regularly used erythromycin for bronchiectasis, the second used amoxicillin for approximately 10 days due to pyelonephritis, and the third regularly used rifaximin for hyperammonemia. In the non-responder group, one individual used amoxicillin for three days due to dental treatment. It is generally believed that the return to the original gut microbiota after antibiotic use takes place within about a month. Both individuals who used amoxicillin did so for a short period, so the impact is considered limited. Moreover, since erythromycin was being used regularly, it is presumed that the gut microbiota of that patient was somewhat stabilized. There were also reports suggesting that rifaximin had no effect on diversity and that the use of antibiotics prior to administration did not affect prognosis in ICI for hepatocellular carcinoma, and since no difference in diversity was observed in the study group, the conclusion was that the effect was minimal. We add the following statement to the result part. (page 5 line160 to 165)
“During the month prior to treatment, there were three patients in the responder group who had used antibiotics: the first patient was on erythromycin for bronchiectasis, the second patient took amoxicillin for approximately 10 days due to pyelonephritis, and the third patient regularly used rifaximin due to hyperammonemia. In the non-responder group, there was one patient who used amoxicillin for three days due to dental treatment.”
Kaji K, Takaya H, Saikawa S, et al. Rifaximin ameliorates hepatic encephalopathy and endotoxemia without affecting the gut microbiome diversity. World J Gastroenterol. 2017;23(47):8355-8366. doi:10.3748/wjg.v23.i47.8355
Zhang L, Chen C, Chai D, et al. The association between antibiotic use and outcomes of HCC patients treated with immune checkpoint inhibitors. Front Immunol. 2022;13:956533. Published 2022 Aug 17. doi:10.3389/fimmu.2022.956533
- The authors mentioned that some patients in the study underwent endoscopic treatment for varices. However, there is no specific data or related discussion on the number of patients who underwent endoscopic therapy in each group, and this difference could affect the overall survival between groups explored in the study.
Thank you for this important comment. We apologize for the insufficient explanation. Among the responder group, 11 patients were noted to have varices before receiving atezolizumab plus bevacizumab, while in the non-responder group, there were 3 individuals. Only one person in each group received preventive endoscopic treatment before administration. No patients encountered difficulties in treatment continuation due to variceal rupture during the administration period. Therefore, we believe there was no impact on prognosis. In view of the above, we would like to add the following expression to result part. (page 5 line 166 to 171)
The presence of varices before the administration of atezolizumab and bevacizumab was observed in 11 patients in the responder group and 3 individuals in the non-responder group. Preventive endoscopic treatment prior to administration was administered to one patient in each group. No patients experienced difficulties in treatment continuation due to variceal rupture during the administration period.
- It is recommended to supplement the references to illustrate the research achievements in this field in recent years. Additionally, in light of the results of this study, it would be beneficial to analyze the current possible confounding or interfering factors.
Thank you for your important remarks. We have added the literature.
Also, as you pointed out, previous reports have indicated that various factors such as BMI, gender, age, race, alcohol intake, quality of bowel movements, dietary habits, etc., can be a factor for going out, but we could not do so due to insufficient statistical power to do factor analysis because of the small sample size. We intend to increase the number of cases and examine these factors in the future. This time, we are planning to modify the discussion section as follows (page 9 line 300 to 303).
Further studies with larger patient cohorts or validation groups are needed to address confounding factors and interfering factors related to the gut microbiome (e.g., race, alcohol intake, quality of defecation, dietary habits).

Reviewer 2 Report
Comments and Suggestions for Authors
Good review
I would recommend if you can also discuss association of gut microbiome and liver cancers.
Ma J, Li J, Jin C, Yang J, Zheng C, Chen K, Xie Y, Yang Y, Bo Z, Wang J, Su Q, Wang J, Chen G, Wang Y. Association of gut microbiome and primary liver cancer: A two-sample Mendelian randomization and case-control study. Liver Int. 2023 Jan;43(1):221-233. doi: 10.1111/liv.15466. Epub 2022 Nov 8. PMID: 36300678.
Schwabe RF, Greten TF. Gut microbiome in HCC - Mechanisms, diagnosis and therapy. J Hepatol. 2020 Feb;72(2):230-238. doi: 10.1016/j.jhep.2019.08.016. PMID: 31954488.
Author Response
Good review
I would recommend if you can also discuss association of gut microbiome and liver cancers.
Ma J, Li J, Jin C, Yang J, Zheng C, Chen K, Xie Y, Yang Y, Bo Z, Wang J, Su Q, Wang J, Chen G, Wang Y. Association of gut microbiome and primary liver cancer: A two-sample Mendelian randomization and case-control study. Liver Int. 2023 Jan;43(1):221-233. doi: 10.1111/liv.15466. Epub 2022 Nov 8. PMID: 36300678.
Schwabe RF, Greten TF. Gut microbiome in HCC - Mechanisms, diagnosis and therapy. J Hepatol. 2020 Feb;72(2):230-238. doi: 10.1016/j.jhep.2019.08.016. PMID: 31954488.
Thank you for important suggestion. I added a sentence on the relationship between hepatocellular carcinoma and intestinal bacteria in the introduction and discussion sections, citing the references you provided (page 2 line45 to 49 and page 8 line 251 to 266).
Added text
The liver and the intestinal tract are connected through the portal vein, and as the gut-liver axis, it is often reported to be closely related to the progression of liver disease and the carcinogenesis and progression of HCC(13, 14). There have been many reports of differences in specific bacteria, and recently Jun et al. reported that Ruminococcaceae, Porphyromonadaceae and Bacteroidetes are associated with a decreased risk of HCC(15). (page 2 line45 to 49)
Through the "gut-liver axis," a bidirectional communication pathway between the gut and the liver, intestinal bacteria and their metabolites exert profound effects on liver health. Dysbiosis, an imbalance in the gut microbiota composition, is associated with various liv-er conditions, such as metabolic-associated steatotic liver disease, alcoholic liver disease, and ultimately, HCC (13, 14).
Many studies have reported differences in gut microbiota in patients with HCC in animal and clinical studies, and research is underway to determine the mechanisms un-derlying these differences. (32, 33). Several key metabolites produced by intestinal bacteria influence liver function and disease progression. These include short-chain fatty acids (SCFAs), bile acids and lipopolysaccharides (LPS)(34). Dysbiosis can disrupt the produc-tion and balance of these metabolites, leading to detrimental effects on liver metabolism and immune response regulation.
The chronic inflammation and liver fibrosis that often precede HCC development are promoted by dysbiosis-induced inflammation and genotoxicity. These factors can lead to DNA damage, genomic instability, and dysregulated cell proliferation and apoptosis pathways, all of which are involved in hepatocarcinogenesis(14). (page 8 line 251 to 266)

Reviewer 3 Report
Comments and Suggestions for Authors
An important role for intestinal bacteria in hepatocellular carcinoma (HCC) has been investigated for about a decade. The intestinal microbiome plays roles not only in the inflammatory process in the liver, but also metabolic, endocrine, and others. Therefore this topic is not new, but the present study does not cite many of the best cited reviews from 2017 and 2020 and both highly cited(https://www.nature.com/articles/nrgastro.2017.72 and https://www.journal-of-hepatology.eu/article/S0168-8278(19)30483-0/fulltext) nor four out of a number of recent 2023 publications (https://www.ncbi.nlm.nih.gov/pmc/articles/PMC10442465/ and https://www.ncbi.nlm.nih.gov/pmc/articles/PMC9109080/ and https://www.mdpi.com/2624-5647/5/2/13 and https://www.nature.com/articles/s41467-022-31312-5) this topic. Specifically it is known that the intestinal microbiome participates and modulates the activities of a number of antibodies used to treat HCC. Thus one major needs in a revised manuscript is not simply citation to these studies but also much more mechanism how bacteria play a role. Important and helpful details of how the intestinal microbiome modulates specific steps to modulate actions of the therapeutic antibodies are well provided in the cited as well as other publications. The focus of the present studies, which are correctly performed and data well presented, is the sole question for the microbiome on response to the specific antibody combination used. This question is important and interesting,, but the science behind the response by the two bacteria most focused on is never disussed. In addition to greater background and mechanisms in Introduction, the manuscript needs to better describe why a study similar to some of the past is needed. Specific populations, specific therapeutic intervention not studied before is described but there are other reasons why this study is important enough to be published.
Comments on the Quality of English Language
minor changes needed
Author Response
Reviewer 3
An important role for intestinal bacteria in hepatocellular carcinoma (HCC) has been investigated for about a decade. The intestinal microbiome plays roles not only in the inflammatory process in the liver, but also metabolic, endocrine, and others. Therefore this topic is not new, but the present study does not cite many of the best cited reviews from 2017 and 2020 and both highly cited(https://www.nature.com/articles/nrgastro.2017.72 and https://www.journal-of-hepatology.eu/article/S0168-8278(19)30483-0/fulltext) nor four out of a number of recent 2023 publications (https://www.ncbi.nlm.nih.gov/pmc/articles/PMC10442465/ and https://www.ncbi.nlm.nih.gov/pmc/articles/PMC9109080/ and https://www.mdpi.com/2624-5647/5/2/13 and https://www.nature.com/articles/s41467-022-31312-5) this topic. Specifically it is known that the intestinal microbiome participates and modulates the activities of a number of antibodies used to treat HCC. Thus one major needs in a revised manuscript is not simply citation to these studies but also much more mechanism how bacteria play a role. Important and helpful details of how the intestinal microbiome modulates specific steps to modulate actions of the therapeutic antibodies are well provided in the cited as well as other publications. The focus of the present studies, which are correctly performed and data well presented, is the sole question for the microbiome on response to the specific antibody combination used. This question is important and interesting,, but the science behind the response by the two bacteria most focused on is never disussed. In addition to greater background and mechanisms in Introduction, the manuscript needs to better describe why a study similar to some of the past is needed. Specific populations, specific therapeutic intervention not studied before is described but there are other reasons why this study is important enough to be published.
Thank you for important point. Citing the references you provided, I have added the following discussion section on possible mechanisms for the results of this study.
Parabacteroides merdae has been reported to potentiate the effects of ICI by being involved in decreasing regulatory T cells, increasing the frequency of Batf3-linage dendritic cells and greater T helper cell 1 responses(36). In a study comparing the combination treatment of anti PD-1 therapy and FMT with anti PD-1 therapy alone in a mouse model of colorectal cancer, the combination treatment was more effective than anti PD-1 therapy alone, and Bacteroides stercoris was one of the intestinal bacteria involved in the metabolic changes in the combination group, with a higher abundance in the combination group(37). Therefore, it is possible that in our study, these bacteria may have been involved in both the immune function and metabolic products of the patients and increased the effect of atezolizumab and bevacizumab.

Round 2
Reviewer 1 Report
Comments and Suggestions for Authors
Thank you to the authors for submitting the revised manuscript. The author's response to the previous review comments is acceptable. We look forward to your publication of subsequent research with a larger sample size.
Comments on the Quality of English LanguageMinor editing of English language required